# Primary health care interventions targeting diabetes, hypertension or dyslipidemia in Malaysia: A scoping review

Xin Rou Teh[1]*, Swee Hung Ang[1], Pei Jia Lee[1], Muhammad Izzuddin Mohd Ropidi[1], Azah  Abdul Samad[2], Sheamini Sivasampu[3], Kim Sui Wan[4]

**1** Institute for Clinical Research, National Institutes of Health, Ministry of Health, Selangor, Malaysia, **2** Family Health Development Division, Ministry of Health, Putrajaya, Malaysia, **3** National Public Health Laboratory, Ministry of Health, Selangor, Malaysia, **4** Institute for Public Health, National Institutes of Health, Ministry of Health, Selangor, Malaysia

* xinrou1801@gmail.com

## Abstract

### Introduction

Despite the high prevalence of diabetes, hypertension and dyslipidemia among Malaysian adults, there are gaps in management and control of these diseases. Evidence suggests that implementation of the Chronic Care Model in primary health care (PHC) can improve patients' clinical outcomes, quality of life and reduce the overall social burden. This study aims to describe the PHC interventions for diabetes, hypertension and/or dyslipidemia in Malaysia and to identify existing gaps by mapping against Chronic Care Model domains.

### Methods

This study reports a section of a larger scoping review and focuses on studies with interventions. PubMed, Embase, Scopus and MyMedR were searched systematically from inception until 31 December 2024, using keywords pertaining to "diabetes", "hypertension", "dyslipidemia", "PHC" and "Malaysia". Study selection was independently performed by reviewers in pairs.

### Results

A total of 32 interventions were identified across 39 publications. The earliest study was published in 2012 and the highest number of publications was seen in 2020. Most studies were conducted in the states of Kelantan and Selangor. The two most common components of intervention were patient education (n = 16) and the use of decision aids (n = 11). Interventions predominantly targeted type 2 diabetes (72%) and the Chronic Care Model domains of self-management support and delivery system design, with very few addressing community linkages (n = 3). Intermediate

**Data availability statement:** All relevant data are within the manuscript and its Supporting information files.

**Funding:** This study was supported by a Malaysia Research Grant from the Ministry of Health Malaysia (NMRR-24-00853-YHT). The funder had no role in study design, data collection and analysis, decision to publish, or preparation of the manuscript.

**Competing interests:** The authors have declared that no competing interests exist.

clinical outcomes (HbA1c, blood pressure, and cholesterol) were the most common measures.

## Discussion/conclusions

This review highlights key gaps in PHC interventions for these three chronic diseases. While self-management and delivery systems are well-addressed, current efforts remain heavily focused on individuals with diabetes, with limited attention to community components and rural populations. There is a need to broaden the intervention scope beyond diabetes and invest in stronger community linkages for a more equitable system in Malaysia.

## Introduction

Cardiovascular disease (CVD) is the leading cause of mortality globally, reaching 19.8 million deaths in 2022 [1]. Diabetes, hypertension and dyslipidemia are the major modifiable risk factors that contribute substantially to this burden. The prevalence of these three non-communicable diseases (NCDs) continues to rise. Global estimates indicated that 1.3 billion adults were living with hypertension in 2019 [2] and 589 million adults had diabetes in 2024 [3], with 80% of the patients living in low- and middle-income countries.

The latest National Health and Morbidity Survey (NHMS) showed that 15.6% of the adult Malaysian population had diabetes, 29.2% had hypertension and 33.3% had high cholesterol [4]. Malaysia's primary health care (PHC) system is a dual-sector system with both public and private service providers. Notably, public and private clinics together handle 85.8% patients with diabetes and 84.2% patients with hypertension [4]. PHC is a whole-of-society approach that can help to strengthen national health systems and ensure health and wellbeing services are brought closer to the community [5]. Hence, enhancing the comprehensiveness and effectiveness of patient care in the PHC system remains a key priority.

The Malaysian healthcare system is continuously striving to combat NCDs. The National Strategic Plan for NCD (NSP-NCD) 2016–2025 includes objectives such as strengthening PHC and setting the national targets to reduce the prevalence of raised blood pressure to 26.0% and halt the rise of the prevalence of diabetes and obesity, maintaining them below 15% by 2025 [6]. In 2017, the Enhanced Primary Health Care (EnPHC) Initiative was launched in 20 public clinics and has since rapidly expanded to over 100 clinics. The EnPHC initiative is a multi-faceted interventional package including population enrollment, risk profiling, integrated care pathways, audits and organisational change. Additionally, there is a well-known community-based initiative called "Komuniti Sihat Pembina Negara" (KOSPEN) that aims to empower and involve the community by recruiting them as volunteers to carry out interventions that target to reduce NCD risk factors [6]. Despite continuous efforts, the prevalence and disease controls of these conditions in Malaysia remain suboptimal.

Wagner et al introduced the Chronic Care Model (CCM) in 1998 [7], emphasising the need to change the care delivery system, provide self-management support, reorganise team function and practice systems, use evidence-based guidelines, and enhance information systems to provide feedback on performance. The locus of care in the CCM model remains to be the personal physician, supported by the team [7]. There are six domains in CCM: (i) self-management support, (ii) delivery system design, (iii) decision support, (iv) clinical information system, (v) health system organization and (vi) community linkages [7]. The CCM model was shown to be effective in reducing HbA1c and blood pressure among adults with type 2 diabetes mellitus (T2DM) in primary care and the effectiveness increased with the number of CCM domains applied [8]. Another systematic review that focused on patients with cardiometabolic multimorbidity in sub-Saharan Africa showed that CCM interventions lower systolic blood pressure, but have mixed results for HbA1c, depressions, medication adherence and quality of life [9].

Hence, this scoping review aims to systematically describe interventions and programs addressing diabetes, hypertension, and/or dyslipidemia at the PHC level in Malaysia. We will summarise the intervention components, their target levels (patient, provider, and/or system) and map the interventions against the six CCM domains to identify gaps that can inform the enhancement of existing interventions or development of new interventions.

## Materials and methods

Our scoping review was guided by the Preferred Reporting Items for Systematic Reviews and Meta-Analyses extension for scoping reviews (PRISMA-ScR) [10] and the Joanna Briggs Institute (JBI) guidelines [11](see PRISMA-ScR checklist in S1 Table). This study reports part of the results of a larger scoping review. This scoping review seeks to address that gap by systematically mapping the existing literature on the quality of care and interventions/programmes for adults with T2DM, hypertension, and/or dyslipidemia in the PHC setting in Malaysia. This study focused on the research question of "What type of healthcare interventions to address T2DM, hypertension, and/or dyslipidemia have been implemented in Malaysia?"

### Search strategy

Four electronic databases (MEDLINE, Scopus, EMBASE, and MyMedR) were searched from inception until 31 December 2024. We developed the search strategy based on the following concept: (diabetes OR hypertension OR dyslipidemia) AND primary health care AND Malaysia. The detailed search strategy can be found in S2 Table. We did not apply any language restriction. The search results were inputted into the Rayyan web-based review management tool [12] for deduplication and further screening by reviewers. We attempted to contact the corresponding authors for the full text if the paper was not available.

### Selection of studies

Two reviewers (X.R.T. & M.I.M.R.) independently reviewed the titles and abstracts, followed by full text screening, based on the inclusion and exclusion criteria. Any disagreements were resolved after discussion or involvement of the third reviewer (S.H.A.). A snowballing method was included where reference lists of the included studies were screened by four reviewers (X.R.T., M.I.M.R., S.H.A. and P.J.L.) to identify possible relevant studies that might have been missed in the database search. In addition, for any study protocols identified during the search, we attempted to locate the corresponding results publication, where available.

Inclusion criteria for the scoping review were studies or programs conducted among adults with T2DM, hypertension or dyslipidemia, or their healthcare providers, in the PHC setting and in Malaysia. Guidelines, book chapters, reviews, study protocols, commentaries, editorials, conference abstracts and letters to editors were excluded. This paper focused on the studies involving the implementation of interventions. Studies that solely described the development of an intervention were excluded.

## Data charting and synthesis

Data extraction was done using a pilot-tested form created in Google Sheet. Three reviewers (X.R.T., M.I.M.R., and P.J.L.) extracted the data using the form, and a random cross-check was performed to ensure data validity. Data items that were extracted included the title, first author's name, study objectives, publication year, study design, study location (state), study population, mean age/age groups, study duration, intervention name, intervention period, brief description of the intervention and outcome measures. The data were tabulated in a narrative way to describe the components of the interventions and the outcome measures used.

We adapted the inductive approach of Intervention Component Analysis to categorize the interventions based on their nature into different components using the steps of reflexive thematic analysis methods by Braun and Clarke [13]. Two reviewers (X.R.T. and S.H.A.) independently examined the interventions and systematically classified them into 14 components, recognizing that each intervention could encompass multiple components. Each reviewer independently identified the intervention component(s) in each of the intervention, followed by a discussion to achieve consensus. The interventions were also categorized according to whether they were targeted at the patient, provider, and/or system level. Lastly, the interventions were further examined for how they align with the six domains of the CCM following the approach outlined above.

## Research registration & ethics approval

This study has been registered with the National Medical Research Register (NMRR-24–00853-YHT) and obtained ethics approval from the Medical Research & Ethics Committee (MREC). The protocol was registered on Open Science Framework (OSF) with the associated project osf.io/gh7b4 and registration DOI: https://doi.org/10.17605/OSF.IO/7GNYM.

## Results

Searches on the four databases yielded 1,778 articles. An additional 108 potential articles and two websites were identified through snowballing of reference lists. After applying the inclusion and exclusion criteria, 371 articles were deemed eligible for the main scoping review (Fig 1). Of these, 39 articles focusing on interventions form the basis of this study. Among the 39 included articles involving intervention, there were a total 32 distinct interventions.

Table 1 provides a summary of the study characteristics, while Table 2 outlines the study characteristics in more detail, such as study location, study design, intervention name, intervention period, a brief description of the intervention, intervention components, and the outcome measures. These articles were published between 2012 and 2023. The number of publications per year ranged from one article in 2017 to ten in 2020 (Fig 2)

### Study design characteristics

The included articles had different study designs, with the top three being randomised controlled trials (RCTs) (n = 20, 51.3%), followed by quasi-experimental studies (n = 7, 17.9%) and cross-sectional studies (n = 5, 12.8%). Most of the RCTs focused on T2DM patients (n = 16, 41.0%), and two studies focused on dyslipidemia and hypertension, respectively.

The sample size of the included studies ranged from 22 to 12,017 participants. When categorized, the majority (n = 19, 48.7%) fell within the 101–500 range. The sample sizes of the included RCTs ranged from 59 to 888 participants. The study with the highest sample size was a quasi-experimental study.

### Study location

When examining the study locations, Selangor emerged as the most frequently included state, appearing in 12 studies (30.8%). This was followed by Kuala Lumpur, Kelantan, and Johor, each represented in six studies. It is important to note that some studies were conducted across multiple states, resulting in overlaps in the location count.

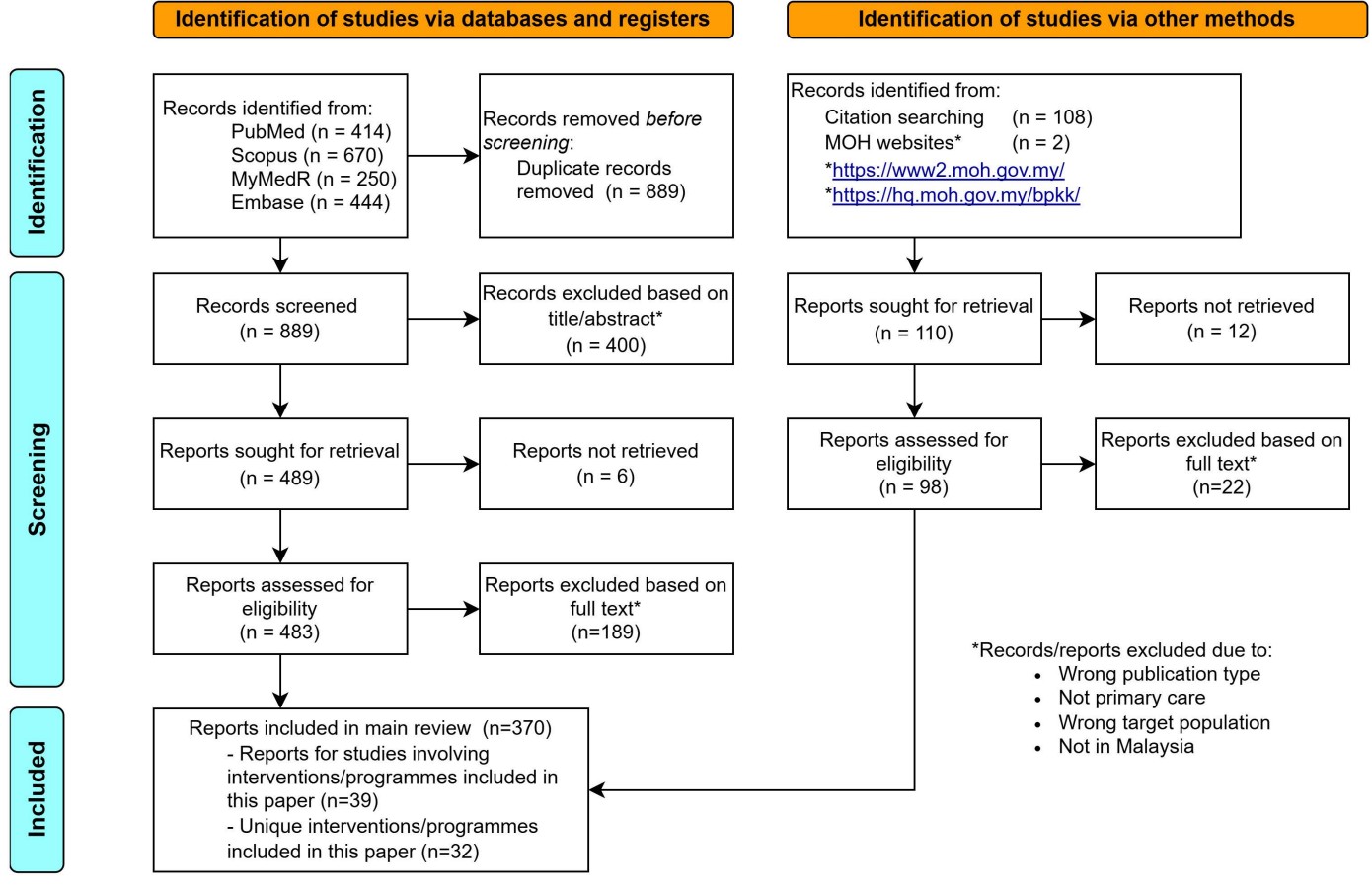

**Fig 1. PRISMA Flow Chart.**

## Study population & study duration

Majority of studies targeted patients with diabetes (n = 28, 71.8%) and a small number focused on healthcare providers (n = 4, 10.3%). In addition, three studies (7.7%) were focusing on hypertension, two studies (5.1%) on dyslipidemia, and two studies (5.1%) with mixed populations, respectively. The mean age of the patients in most studies was 50 years or older. Whereas the mean age for healthcare providers ranged from 30–40 years old. The duration of intervention ranged from a single session to two years, with nearly half of the interventions lasting between one to six months (n = 19, 48.7%). Some studies evaluated existing interventions and therefore the duration was not specified.

## Intervention components

**Summary of intervention components.** The intervention components were highlighted in Table 2. Of the 32 interventions reviewed, ten interventions were multifaceted and included two or more distinct components or implementation strategies. For example, the EnPHC intervention incorporated seven distinct components, such as Family Doctor Concept (FDC), medication therapy adherence clinic (MTAC), care coordinator, audits and feedback, multidisciplinary team, decision aid, and patient management system.

**(i) Education (n=16).** Education was the most implemented component across interventions, identified in 16 articles. We further categorised it into focused topic education and comprehensive education. Focused interventions addressed

**Table 1. Study Characteristics.**

| Characteristics | Number of articles, n | Percentage, % |
|---|---|---|
| *Study design* | | |
| Randomized controlled trial | 20 | 51.3 |
| Quasi-experimental study | 7 | 17.9 |
| Cross-sectional study | 5 | 12.8 |
| Pre/Post study | 4 | 10.3 |
| Mixed-method study | 2 | 5.1 |
| Comparator study | 1 | 2.6 |
| *Study Location (States)* | | |
| Kedah | 1 | 2.6 |
| Kelantan | 6 | 15.4 |
| Klang Valley | 1 | 2.6 |
| Johor | 4 | 10.3 |
| Negeri Sembilan | 3 | 7.7 |
| Perak | 2 | 5.1 |
| Pulau Pinang | 2 | 5.1 |
| Sabah | 1 | 2.6 |
| Sarawak | 1 | 2.6 |
| Selangor | 5 | 12.8 |
| Selangor/Kuala Lumpur | 4 | 10.3 |
| Selangor/Johor | 2 | 5.1 |
| Kuala Lumpur/Putrajaya | 2 | 5.1 |
| Selangor/Negeri Sembilan | 1 | 2.6 |
| Peninsular Malaysia | 1 | 2.6 |
| Malaysia (not specifying the states) | 3 | 7.7 |
| *Sample size* | | |
| <50 | 3 | 7.7 |
| 50–99 | 7 | 17.9 |
| 100–499 | 19 | 48.7 |
| 500–999 | 6 | 15.4 |
| ≥1000 | 4 | 10.3 |
| *Study population* | | |
| Healthcare providers | 4 | 10.3 |
| Patients with diabetes | 28 | 71.8 |
| Patients with hypertension | 3 | 7.7 |
| Patients with dyslipidemia | 2 | 5.1 |
| Mixed patient population (diabetes, hypertension or dyslipidemia) | 2 | 5.1 |

* There were studies with the same intervention.

specific areas such as insulin injection technique [20,35], education on insulin [16], erectile dysfunction [15], sharp disposal [31], wound care [52], men's health [49], and self-monitoring of blood glucose (SMBG) [39]. Comprehensive education included structured programs covering broader topics, including home-based education [38], group nutrition counselling [45], multidisciplinary healthy lifestyle course [21], comprehensive training module on diabetes care for providers [51] and patients [29,30], as well as multi-day workshops on emotional, social support and goal setting [32,33].

**Table 2. Characteristics of the included studies (n = 39).**

| First author (publication year) & Study Design | Study population (n) & study location (state(s)) | Intervention name & intervention period | Brief description of the intervention & intervention component* | Outcome measures | Results (Primary Outcome) |
|---|---|---|---|---|---|
| **T2DM (n = 28)** | | | | | |
| **Husin M (2023) [14]** Quasi-experimental study | T2DM (n = 12,017) Study location(s): Selangor/Johor | EnPHC • 2 years (July 2017- June 2019) | EnPHC interventions include: i) an Integrated Care Pathway ii) a patient visit checklist iii) Integrated Specialized Services by allied health professionals iv) NCD screening and cardiovascular risk stratification v) an NCD care form vi) the Family Health Teams concept vii) involvement of a care coordinator viii) pharmacist-led Cardiovascular Care Bundle Medication Therapy Adherence Clinic ix) clinical and prescribing audits x) structured communication across primary and secondary care levels with a fast-track referral system  **Intervention component*: 3, 4, 5, 6, 9, 10, 14** | 1. Biomarkers (HbA1c, BP, LDL, HDL) 2. Processes – test done (HbA1c, blood glucose, lipid test, LDL, urine microalbumin, liver function test, BP, BMI, fundus examination, foot examination, CVD risk assessment) 3. Counselling provided (exercise, diet) 4. Prescription (lipid lowering drug) | 1. Biomarkers: No change in HbA1c, BP, LDL and HDL control 2. Process of care measures: Improved for HbA1c/lipid/LDL tests, Urine albumin, BMI measurement, CV risk assessment, and exercise counselling. No change for fundus/foot examinations, LFT, diet counseling, BP measurement and use of lipid-lowering drugs. |
| **Muin MRA (2023) [15]** Quasi-experimental study | T2DM (n = 120) Study location(s): Kedah | Prompt sheet for ED discussion and "Knowledge Translation Tools in the Management of Erectile Dysfunction" (LASTED) • Single session | The prompt sheet contained brief information on ED and options for ED discussion. The 4 options were: "I do not want to discuss ED;" "I want to discuss the risk of ED;" "I want to discuss treatment of ED;" and "I want to discuss the severity of ED." The patients were asked to give the completed prompt sheet to their physician at the beginning of the consultation. The consultation then proceeded with the physician using the LASTED flipchart based on the options selected on the prompt sheet.  **Intervention component*: 1, 10** | 1. Total score of 5-item International Index of Erectile Function (IIEF-5) 2. Initiation of ED discussion 3. Prescription of PDE5 inhibitors 4. Use of LASTED flipchart 5. Satisfaction with the consultation | 1. Increased in initiation of ED discussion 2. Higher number of PDE5 inhibitors prescription |
| **Mohd Tahir NS (2023) [16]** RCT | T2DM (n = 88) Study location(s): Kelantan | Malay version Diabetes Conversation Maps (DCM) • Single session | - DCM topic was on insulin treatment initiation. - The groups consist of 8–10 participants with one trained facilitator. Each session is about 60–90 minutes. The facilitator explored their baseline knowledge and their attitudes towards diabetes. The facilitator will read out loud the myth cards to stimulate a discussion. She then presented the facts cards in a clear, proper and concise way.  **Intervention component*: 1** | 1. Insulin acceptance | Increased insulin acceptance among patients who initially refused insulin treatment |

*(Continued)*

| First author (publication year) & Study Design | Study population (n) & study location (state(s)) | Intervention name & intervention period | Brief description of the intervention & intervention component* | Outcome measures | Results (Primary Outcome) |
|---|---|---|---|---|---|
| **Lim PC (2023) [17]** Pre/Post study | Diabetic (n = 956) Study location(s): Pulau Pinang | Diabetes Medication Therapy Adherence Clinic (DMTAC) • (existing program) – Lim PC (2023) & Lee XY (2015) • 12 months – Chai A (2020) | - Pharmacist incorporated the MTAC Diabetes Protocols in managing the medication and disease<br>- Medication initiation or discontinuations, dosage adjustments, and recommendations for laboratory investigations.<br>- Pharmacists were granted authorization by the physicians to carry out insulin dosage adjustments<br>- DMTAC pharmacists routinely documented patients' medical, social, and family histories, drug-related issues, interventions, laboratory data, treatment regimens, and adherence using the Malaysian Medication Adherence Tool (MyMAAT) score<br>- Patient was required to attend at least four DMTAC visits.<br><br>**Intervention component*: 4** | 1. HbA1c | HbA1c reduced significantly from baseline to post 1 (4–6 months) and post 2 (8–12 months). |
| **Chai A (2020) [18]** RCT | T2DM (n = 100) Study location(s): Sarawak | | | 1. Biomarkers (HbA1c, BMI) 2. Medication compliance | HbA1c reduction at 12th month compared to baseline |
| **Lee XY (2015) [19]** Pre/Post study | T2DM (n = 56) Study location(s): Kuala Lumpur/ Putrajaya | | | 1. HbA1c 2. Medication understanding score 3. Medication adherence | 1. Mean HbA1c reduced at post-4th session. 2. The mean medication understanding score was significantly higher than the score at baseline. 3. The mean adherence score was higher than baseline. |
| **Selvadurai S (2021) [20]** RCT | T2DM (n = 160) Study location(s): Kuala Lumpur/ Putrajaya | Insulin injection technique re-education • 4 months | Patients were given intensive re-education during their monthly medication acquisition visits for 4 months. Each patient was given comprehensive education on their injection technique knowledge technique including LH physical examination and received an education kit consisting of a site rotation grid, educational insulin injection technique and LH leaflet during recruitment.<br><br>**Intervention component*: 1, 10** | 1. Insulin injection technique 2. Medication adherence 3. Perception on diabetes and treatment 4. Improvement of lipohypertrophy presence 5. HbA1c | There were significant improvements in insulin injection technique, medication adherence, and patient perception on diabetes and treatment. |
| **Tay CL (2021) [21]** Comparator study | T2DM (n = 220) Study location(s): Perak | Healthy Lifestyle Education (HLE) course • Single session | Healthy Lifestyle Education (HLE) is a structured group educational 3-hour course. The course is conducted by six different healthcare providers (doctor, pharmacist, medical assistant, nutritionist, occupational therapist and physiotherapist); with 30-minute allocation for each topic.<br><br>**Intervention component*: 1** | 1. Biomarkers (Weight, SBP, DBP, HbA1c, fasting blood sugar, TC, TG, LDL, HbA1c reduction) 2. Prescription (insulin's total daily dose) | There were significant improvements in the median of HbA1c, FBS and TC. No changes in terms of weight, SBP, DBP, TG, LDL and insulin's total daily dose. |
| **Nordin N (2021) [22]** Cross-sectional study | T2DM (n = 772) Study location(s): Kelantan | Family Doctor Concept (FDC) • (existing program) | - The FDC restructures the primary healthcare services, whereby patients and population are taken care of by a specific primary healthcare team (PHCT) according to zone.<br>- "One Family One Doctor" concept<br>- The major differences between FDC and non-FDC clinics are the availability of equipment (e.g., fundus camera and X-ray modality) and specialty services (e.g., FMS, radiology, laboratory, nutrition and diet, physiotherapy, and occupational therapy) that are more available at FDC clinics.<br><br>**Intervention component*: 3** | 1. Satisfaction of doctor-patient interaction (using SKIP-11 questionnaire) 2. Perceived quality of care (using PACIC-M questionnaire) | 1. No difference in perceived quality of care between FDC and non-FDC clinics. 2. Higher satisfaction of doctor-patient interaction in FDC clinics. |
| **Nordin N (2020) [23]** Cross-sectional study | | | | 1. HbA1c ≤ 6.5% 2. *Skala Kepuasan Interaksi Perubatan-11* (SKIP-11): -Satisfied doctor-patient interaction -Satisfied distress relief -Satisfied rapport -Good interaction outcome 3. Number of patients and healthcare professionals (medical officers, diabetic educators, pharmacists, dieticians, physiotherapists) 4. Availability of equipments (X-ray and fundus camera) | 1. Overall satisfaction did not show a significant difference. 2. Subdomain distress relief and interaction outcome show a significant difference, with patients attending FDC clinics having higher satisfaction. 3. FDC clinics have better glycemic control |

*(Continued)*

| First author (publication year) & Study Design | Study population (n) & study location (state(s)) | Intervention name & intervention period | Brief description of the intervention & intervention component* | Outcome measures | Results (Primary Outcome) |
|---|---|---|---|---|---|
| **Ngah NF (2020) [24]** Cross-sectional study | DM (n = 3305) <u>Study location(s):</u> Selangor | Retinal Disease Awareness Program (RDAP) • (existing program) | - Retinal Disease Awareness Program (RDAP) was initiated to enhance the existing screening programs conducted by the local health clinics. <br>- The eye examination was conducted by the ophthalmology team with visual acuity testing and fundus photography for diabetic retinopathy (DR) assessment. <br>- A trained photographer took retinal images and sent them to a remote trained reader (typically an ophthalmologist or optometrist) for interpretation. <br><br>*Intervention component\*: 7* | 1. Prevalence of DR | 9% patients were found to have DR and other visual complications such as maculopathy (0.9%), cataract (4.8%) and glaucoma (0.4%). |
| **Tajudin TR (2020) [25]** Cross-sectional study | DM (n = 22) <u>Study location(s):</u> Johor | Diabetes Awareness Day & diabetes clinic (primary care-based behaviour modification program) • 6 months | - During the diabetes awareness day, participants were offered free blood tests. Breakfast and lunch meals were provided and prepared by a caterer with a strong background of diabetes food preparation. There were several talks on diabetes and its management. <br>- Diabetes clinic is held once a week with a patient load of five to seven patients per session. Patients were followed up every month for diabetic treatment and management for a total of six months. Each follow-up session has at least half an hour consultation to allow adequate physical examination, review of laboratory results and discussion of problems faced by the patients. <br><br>*Intervention component\*: 7, 8* | 1. Biomarkers (FBS, HbA1c, LDL, HDL, TG) | FBS and HbA1c did not show any significant difference pre- and post-intervention. |
| **Daud MH (2020) [26]** RCT | T2DM (n = 888) <u>Study location(s):</u> Selangor/ Kuala Lumpur | EMPOWER-PAR • 12 months | - Formation and training of the Chronic Disease Management (CDM) Team <br>- Distribution and utilisation of the intervention tools: Malaysian CPG and the Quick References (QR) on the Management of T2DM and the Global CV Risks Self-Management Booklet <br>- Facilitation and support to implement the intervention <br><br>*Intervention component\*: 9, 10* | 1. Processes recorded (BP, smoking status, BMI, WC, family history of premature CVD) <br>2. Processes done (Foot examination, FSL, HbA1c, fundus, ECG, RP, urine protein, lifestyle modification and self-management advice given) <br>3. Interval between follow up visits did not exceed 6 months | 1. There were significant improvements in the absolute change in the proportion of PCP's adherence for recording of smoking status, screening of erectile dysfunction among males age > 40 and BMI; performing foot examination; FSL measurement; performing funduscopy/fundus photography; performing ECG; monitoring of renal profile; urine protein measurement, and giving lifestyle modification and self-management advice in the intervention group compared to the control group at 1-year follow-up. <br>2. There was no change in terms of HbA1c measurement. <br>3. Family history of premature CVD, BP recording and follow up interval were all 100% throughout. |
| **Ramli AS (2016) [27]** RCT | | | | 1. Biomarkers achieving target (HbA1c, BP, WC, BMI, TC, TG, LDL, HDL) | The intervention group was twice more likely to achieve the HbA1c target compared to those in the control group. |

*(Continued)*

**Table 2.** (Continued)

| First author (publication year) & Study Design | Study population (n) & study location (state(s)) | Intervention name & intervention period | Brief description of the intervention & intervention component* | Outcome measures | Results (Primary Outcome) |
|---|---|---|---|---|---|
| Lee JY (2020) [28] RCT | T2DM (n = 240) Study location(s): Selangor | Glucose telemonitoring • • 6 months | - Participants are provided a gluco-telemeter and were instructed to transmit up to 6 glucose readings weekly, which will be uploaded to a central server. - Participants received automated feedback on their glycemic and metabolic results. - Monthly communications from the research team on self-management skills, blood glucose control, and the importance of medication adherence aimed at educating and motivating patients. - Clinic visits at weeks 4, 12, and 24, where additional diabetes self-management education was given.  _Intervention component*: 2_ | 1. Biomarkers (HbA1c, fasting plasma glucose, TC, TG, LDL, HDL, SBP, DBP, achieve HbA1c<7%) 2. Diabetes knowledge test 3. Emotional distress 4. Health-related quality of life (EuroQol-5D) 5. Self-efficacy 6. Self-blood glucose testing 7. Annual direct program cost | There was no significance difference in HbA1c levels at weeks 24 and 52 from baseline visit. |
| Ayadurai S (2020) [29] RCT | T2DM (n = 154) Study location(s): Johor | Simpler Tool • 6 months (27 weeks) | A multifaceted diabetes intervention tool (Simpler tool) – Statin or cholesterol control, insulin or glycemic control, medication, blood pressure, lifestyle, education, and cardiovascular risk prevention strategies. The intervention group were given appointments to meet with the pharmacist trained with Simpler Tool.  _Intervention component*: 1, 10_ | 1. Biomarker (BMI, WC) 2. Medication adherence 3. Quality of life 4. Duration of self-reported exercise | 1.Significant increase in the number of total antiplatelet medicines prescribed at the end of the study in comparison with the UC group. 2.No statistically significant increase in the number of patients prescribed with metformin, insulin or initiation of ACEi or ARB. 3.The most common MRP was patients' non-adherence (44.9%), followed by sub-therapeutic dose (21.6%) and the need for additional therapy (17.3%). 4.The most frequent causes of non-adherence to medicines were patients forgetting to take their medicine (68.1%), followed by patients preferring not to take their medicine (23.1%). 5.The most common interventions provided by pharmacists to improve adherence among SC patients were recommendations to buy a pill reminder device or providing a medicine timing chart (65.0%), followed by reinitiating drug therapy based on patient collaboration (35.0%). 6.The most common suggestion accepted by doctors, and consequent therapy change, was adding a medicine (45.1%). 7.The duration of self-reported exercise significantly increased at six months 8.Patient self-reported medicine adherence improved at six months compared with baseline. 9.The overall QOL improved significantly in both arms at six months. The SC arm had significant improvements at all four domains at six months compared with baseline: physical health; psychological; social relationships; and environment. Two domains (physical health and environment) were significantly improved in the SC arm compared with the UC arm at six months. |
| Ayadurai S (2018) [30] RCT | | | | 1. Biomarkers (HbA1c, SBP, DBP, TC, TG, LDL, HDL) | Significant mean HbA1c reduction of at least 1% was observed in intervention arm |

_(Continued)_

| First author (publication year) & Study Design | Study population (n) & study location (state(s)) | Intervention name & intervention period | Brief description of the intervention & intervention component* | Outcome measures | Results (Primary Outcome) |
|---|---|---|---|---|---|
| **Hasan UA (2019) [31]** Quasi-experimental study | T2DM (n = 136) Study location(s): Kelantan | Locally adapted and validated Diabetes Community Sharp Disposal Education Module • 3 months | The Diabetes Community Sharp Disposal Education Module consisted of four main topics: (1) medical sharps used for treatment of diabetes in the community; (2) proper handling of sharps prior to disposal; (3) improper community sharp disposal methods; (4) proper community sharp disposal method The educational materials were in the form of lectures and demonstrations, flip charts and pamphlets, which were written in the local Malay language. *Intervention component*: 1* | 1. Malay Version of Diabetes Community Sharp Disposal (M-DCSD) knowledge score 2. Community sharp disposal practice | 1.An overall significant higher mean M-DCSD knowledge score. 2.For between-group differences with regards to time, there was significant interaction between the groups and time (one month and three months) in the M-DCSD knowledge score. |
| **Chew BH (2019) [32]** RCT  **Chew BH (2018) [33]** RCT | T2DM (n = 124 for 2018 study and n = 295 for 2019 study) Study location(s): Negeri Sembilan | Value-based emotion-focused educational program (VEMOFIT) • 6 weeks (booster session at week 18) | - Nurses from the participating health clinics are invited to participate in a 2-days training course. - Each participant in the VEMOFIT program was allowed to bring along one significant other person as a co-participant. - The VEMOFIT program included: (1) exploring personal beliefs regarding diabetes; (2) training emotional skills; and (3) providing social support and setting short- and long-term goals. *Intervention component*: 1* | 1. Biomarkers (HbA1c, SBP, DBP, LDL) 2. Diabetes-related distress 3. Depressive symptoms 4. Illness perception on diabetes 5. Quality of life 6. Diabetes self-efficacy 7. Diabetes self-care activities 8. Positive emotions 9. Prescription (OHA, insulin, antihypertensive agents, lipid-lowering agents)** **only in Chew BH (2019)* | 1.The mean DDS-17 level decreased in both groups (with a significantly higher reduction in the attention-control group. 2.Attention control program has a bigger change from baseline, with adjusted intervention effect at 12-month  The adjusted differences between groups for mean distress level in both the intention-to-treat and per protocol analysis were not significant. |
| **Chee WSS (2017) [34]** RCT | T2DM (n = 230) Study location(s): Negeri Sembilan | Malaysian trans-cultural diabetes nutrition algorithm (tDNA) • 6 months | - Patients underwent an initial risk stratification checklist and were prescribed medical nutrition therapy (MNT) consisting of a structured low-calorie meal plan (1200 or 1500 kcal/day) and a physical activity prescription of at least 150 min per week. - tDNA toolkit. - The tDNA-MI subgroup received counselling that incorporated motivational interviewing principles. - The tDNA-CC subgroup received conventional counselling techniques that focused on empathetic listening, education, persuasion, and encouragement. *Intervention component*: 10, 11, 12* | 1. Biomarkers (Weight, BMI, WC, body fat, HbA1c, fasting plasma glucose, SBP, DBP, TC, LDL, HDL, TG, hsCRP) 2. Meal replacement adherence rates 3. Dietary intake (energy intake, carbohydrate, protein, fat) 4. Physical activity level 5. Symptomatic hypoglycemia 6. Prescription (OHA use) | After 6 months: A1c improved significantly for patients in both the tDNA intervention groups, with the tDNA-MI patients achieving a greater lowering of A1c values ($-1.1 \pm 0.1\%$, $p < 0.001$) than the tDNA-CC patients ($-0.5 \pm 0.1\%$, $p < 0.001$). After 12 months (6 months post-intervention): The tDNA-MI group maintained a significant reduction of A1c from baseline during the follow-up phase, but tDNA-CC and UC patients had A1c values returning to baseline ($-0.5 \pm 0.2\%$ vs $0.1 \pm 0.1\%$ vs $0.1 \pm 0.1\%$, $p = 0.007$, respectively. |

*(Continued)*

| First author (publication year) & Study Design | Study population (n) & study location (state(s)) | Intervention name & intervention period | Brief description of the intervention & intervention component* | Outcome measures | Results (Primary Outcome) |
|---|---|---|---|---|---|
| Ahmad S (2016) [35] Pre/Post study | Diabetic (n = 138) Study location(s): Selangor | Insulin injection technique education • 20 minutes | - Education on proper insulin injection technique was delivered using placebo injection device.<br>- Individualization of education<br><br>*Intervention component\*: 1* | 1. HbA1c<br>2. Insulin injection technique appropriateness<br>3. Steps of insulin injection techniques | 1.Patients' insulin injection technique appropriateness increased.<br>2.Steps of insulin injection technique:<br>◦ Check expiry date of cartridge and/or amount of insulin (Pre: 6.1% vs Post: 54.4%)<br>◦ Turn the pen up and down and/or roll the pen 10x (Pre: 50.9% vs Post 91.2%)<br>◦ Dial 2–4 unit to perform air-shots (Pre: 28.1% vs Post: 76.3%)<br>◦ Choose a site for injection (Pre: 57.9% vs Post: 92.1%)<br>◦ Dial required dose (Pre: 100% vs Post: 100%)<br>◦ Pinch the skin (Pre: 45.6% vs Post: 86%)<br>◦ Insert needle smoothly into skin and press plunger until the button stop moving (Pre: 98.2% vs Post:100%)<br>◦ Count to ten before removing needle from skin (dwell time) (Pre: 62.3% vs Post: 93.9%)<br>◦ Check to make sure you see a '0' in the dose window (Pre: 43.9% vs Post: 85.1%)<br>3. Mean HbA1c levels had significant change from 9.9% to 9.08%. |
| Sazlina SG (2015) [36] RCT | T2DM (n = 69) Study location(s): Selangor | Personalized feedback (PF) with or without peer support (PS) • 3 months | - PF received feedback comprised participants' physical activity patterns.<br>- PS group received support from peer mentors<br><br>*Intervention component\*: 12* | 1. Biomarker (Body fat, weight, BMI, WC, HbA1c, BP, LDL, HDL, TG)<br>2. Physical activity level (pedometer and subjective measures using diary and Physical Activity Scale for the Elderly (PASE))<br>3. Cardiovascular risk factors<br>4. Functional status (cardiorespiratory fitness and balance using 6-min walk test and timed up and go test)<br>5. Quality of life<br>6. Psychosocial wellbeing | Greater mean daily pedometer readings in the PS group compared to the PF group. |
| Lee JY (2015) [37] Mixed-method study (Trial with FGD) | T2DM (n = 32) Study location(s): Not specified | Diabetes telemonitoring • 7 weeks | TG participants received a web-enabled glucometer that enables self-monitoring and goal setting. Recorded readings were transmitted to an online portal. Participants received feedback upon receipt of transmission when there were three continuous blood glucose values of ≤3.9 mmol/l and ≥11.1 mmol/l.<br><br>*Intervention component\*: 2* | 1. Biomarkers (fasting blood glucose, serum fructosamine, LDL, HDL, TG, TC)<br>2. Hypoglycemic episodes<br>3. Hypoglycemic symptoms | 1.No changes were observed for biochemical outcomes at the end of the study (FPG, serum fructosamine, LDL cholesterol, HDL cholesterol, Triglyceride, Total cholesterol)<br>2.Intervention group experienced fewer hypoglycemic symptoms during the study period<br>3.Proportion of TG participants experiencing self-reported hypoglycemia was significantly lower |
| Chow EP (2015) [38] RCT | T2DM (n = 150) Study location(s): Pulau Pinang | Pharmacist-led home-based diabetes education • 3 months | - Home visits by a trained pharmacist with home-based educational intervention (proper use of medications and on T2DM)<br>- Patients were given an information booklet on T2DM and a food pyramid chart prepared by the Ministry of Health, Malaysia.<br>- Medication charts and pictorial labels for each of the medications were given when necessary.<br>- Telephone reminders for monthly prescription refills and reassessments of medication use.<br><br>*Intervention component\*: 1, 8, 10* | 1. HbA1c<br>2. Medication adherence<br>3. Diabetes knowledge | 1. Higher knowledge of disease<br>2. Higher diabetes adherence score<br>3. Lower HbA1c level |

*(Continued)*

Table 2. (Continued)

| First author (publication year) & Study Design | Study population (n) & study location (state(s)) | Intervention name & intervention period | Brief description of the intervention & intervention component* | Outcome measures | Results (Primary Outcome) |
|---|---|---|---|---|---|
| **Ismail M (2013) [39]** RCT | T2DM (n = 105) Study location(s): Selangor/ Negeri Sembilan | Self-monitoring blood glucose (SMBG) • 6 months | -Two-day classes that included practical demonstrations of SMBG and the usage of the glucometer was explained. -Patients were supplied with a glucometer with reagent test strips at no charge. -Patients were advised to monitor their blood glucose levels, to keep a record in their logbooks, and adjust the dose of OHA/insulin accordingly. -Required to visit their doctor at intervals of two months and to see the nurse every month to record their SMBG results. *Intervention component\*: 1, 2, 8* | 1. Biomarkers (SBP, DBP, WC, BMI, HbA1c, TC, TG, achieved glycemic control) | 1.Significant reduction in HbA1c levels 2.Higher percentage of DM patients reaching the treatment target (HbA1c ≤ 7%) |
| **Muhamad R (2012) [40]** RCT | T2DM (n = 82) Study location(s): Kelantan | Postprandial and fasting blood glucose monitoring • 6 months | All the subjects underwent the blood glucose monitoring protocols for the duration of 6 months with monthly follow up. There were two blood samples taken: First sample was collected before breakfast (fasting or pre-breakfast) and the second one at 2-hour post breakfast. All patients were served with standard breakfast which consisted of ¾ cup of rice (30g carbohydrate), fish (10g protein) and coconut milk (10g fat). **Intervention component\*: 2** | 1. Biomarker (HbA1c control, fructosamine control) | There was a significantly lower mean HbA1c and mean Fructosamine in the post-prandial group compared to the fasting group and post-prandial group. |
| **Wong SSL (2012) [41]** RCT | T2DM (n = 500) Study location(s): Selangor | Colour-coded HbA1c-graphical record • 6 months | - Colour-coded HbA1c-graphical record which incorporates traffic light signal colours to denote HbA1c ranges. - The HbA1c result was plotted on the graph at each follow-up (x-axis) and the trend was exhibited by drawing a line between the results. *Intervention component\*: 10* | 1. Biomarker (HbA1c) 2. HbA1c knowledge | No significant difference in the improvement of mean HbA1c level and mean change of HbA1c knowledge score, between the two groups at 3 and 6 months. |
| **Dyslipidemia (n = 2)** | | | | | |
| **Heng WK (2019) [42]** RCT | Dyslipidemia (n = 147) Study location(s): Malaysia | Time of simvastatin • 4 months | -The defined time frames for after breakfast is between 6:00 am to 10:00 am; after dinner between 5:00 pm to 9.00 pm; and at bedtime accordingly. *Intervention component\*: 13* | 1. Biomarkers (TC, LDL, HDL, TG) 2. Medication adherence (MMAS) | LDL-C was reduced significantly from baseline at both follow ups in all arms. The differences of LDL-C percentage reduction among three arms were statistically significantly different. |
| **Selvaraj FJ (2012) [43]** RCT | Dyslipidemia (n = 297) Study location(s): Malaysia | COACH program • 36 weeks | - COACH health booklet - bi-weekly telephone follow-up by trained nurse educators for 24 weeks. - phone calls and SMS to remind them about forthcoming follow-up visits *Intervention component\*: 8, 10* | 1. Biomarkers (TC, LDL, HDL, TC:HDL ratio, TG, SBP, DBP) 2. Statin treatment compliance 3. Framingham CHD risk scores 4. Lifestyle modification (i.e., smoking, dietary changes, physical activity) 5. Patient satisfaction | 1.At week 24, LDL-C improvement was greater in the intervention group but not significant (p = 0.288). 2.A significant decline from week 24–36 followed withdrawal of the COACH program. There was no significant difference in lipid outcomes between 2 study groups at week 36 (12 weeks after withdrawal from COACH program) |
| **Hypertension (n = 3)** | | | | | |
| **Norwati D (2023) [44]** RCT | HPT (n = 202) Study location(s): Kelantan | Quran recitation audio • Single session | - Listen to the Quran recitation - a copy of the Quran with Malay (native language) translation *Intervention component\*: 12* | 1. Biomarkers (SBP, DBP, pulse rate) | There were significant differences in SBP and DBP in the intervention group compared to the control group. |

*(Continued)*

**Table 2.** (Continued)

| First author (publication year) & Study Design | Study population (n) & study location (state(s)) | Intervention name & intervention period | Brief description of the intervention & intervention component* | Outcome measures | Results (Primary Outcome) |
|---|---|---|---|---|---|
| **Lee SS (2022) [45]** RCT | HPT (n = 74) <u>Study location(s):</u> Johor | DePEC (Dementia Prevention and Enhanced Care)-Nutrition trial • 8 months | Complex dietary and behavioural intervention evaluating the feasibility of the combination of salt intake reduction and increased high-nitrate vegetable consumption among middle-aged and older Malaysian adults with elevated blood pressure 1. Group nutritional counselling was delivered during the baseline clinic visit by a trained medical officer using PowerPoint slides and practical activities. Each session lasted between 1–1.5 hours. 2. Information booklets contained recipes of low salt and/or high-nitrate meals that participants could prepare at home. The booklet also provided several sources and links for further information. 3. A salt measuring spoon (salt intervention) to support them in understanding portion sizes and measuring salt intake. The measuring spoon had a dual side with nine adjustable scales (0.5, 1, 2, 3.5, 5, 7, 9, 11 and 13 g). Participants were taught how to use the spoon during the group counselling sessions. 4. Biweekly text messages – educational messages and reminders of the key dietary behaviour changes to encourage adherence. 5. Reinforcement video messages to remind participants of the key dietary advice that was discussed during the baseline counselling sessions. *Intervention component*: 1, 10* | 1. Feasibility of the recruitment, follow-up, retention, data collection procedures, clinical outcome measures and interventions 2. Clinical and nutritional measures (blood pressure, Montreal Cognitive Assessment (MoCA) test, auditory verbal learning test (AVLT), 24-h recall, Food Frequency Questionnaire (FFQ), and nitrate concentrations in urine, saliva and blood samples) | 1.Of the 225 eligible participants, 97 (43%) consented, while 23% were unable to arrange the baseline assessment and 26% did not agree to participate. 2.Follow-up attendance rates were 79% for interim 1 (4 months) and 77% for interim 2 (6 months). 3.The median intervention period was 10.5 months. 4.The overall retention rate was 73% and did not differ between study groups. 5.Most participants (96%) provided blood samples and physical measurements during the baseline assessment. 6.The majority of participants indicated the usefulness of group counselling, text messages, videos, and the information booklet. 7.Overall, both low salt and high-nitrate vegetable interventions were well accepted by participants. |
| **Low WHH (2013) [46]** Non-randomized controlled trial (quasi-experimental) | HPT (n = 486) HCP (n = 70) <u>Study location(s):</u> Selangor/ Kuala Lumpur | Community-Based Cardiovascular Risk Factors Intervention Strategies (CORFIS) • 6 months | - The allied health care team was constituted purposefully to support individual GPs. - Each team member delivered care according to an agreed management and drug treatment protocol - A custom designed, secure web-based application was set-up to capture patients' data as well as to organise and coordinate care among all the healthcare providers. - The trained allied health professionals counsel the patient monthly at their assigned GP clinics. - Continuity of care for patients by the same care-provider was emphasised. - Home monitoring devices were loaned to the patients. - Patients were provided with information on local patient associations and support groups. **Intervention component*: 2, 3, 8, 9, 14** | 1. Biomarker (BP control, SBP, DBP) | A higher proportion of participants with hypertension in the intervention group achieved target BP. |

*(Continued)*

 

**Table 2.** (Continued)

| First author (publication year) & Study Design | Study population (n) & study location (state(s)) | Intervention name & intervention period | Brief description of the intervention & intervention component* | Outcome measures | Results (Primary Outcome) |
|---|---|---|---|---|---|
| **Mixed population (n = 2)** | | | | | |
| **Tan LK (2023) [47]** Cross-sectional study | Obesity/ DM/ Hypertension/ Dyslipidemia (n = 9760) Study location(s): Malaysia | Malaysia Healthy Plate (MHP) • (existing program) | -MHP was introduced by the MOH Malaysia in 2016 with the Tagline "Suku suku separuh" (Quarter Quarter Half) - The MHP translates the recommendations of the Malaysian Dietary Guidelines (MDG) and the Malaysian Food Pyramid into a ten-inch round plate that is divided into three portions, i.e., two quarters and a half, as a visual aid to Malaysians to practice healthy eating habits. - The divided round plate serves as a blueprint of the total food in each food group that needs to be consumed in a meal to achieve a healthy and balanced diet. - A healthy plate is comprised of a quarter plate of grains (i.e., rice, other cereals, whole grains, cereal-based products, and tubers), followed by a quarter plate of (i.e., fish, poultry, eggs, meat, and legumes) and a half plate of fruit and vegetables (FV)<br><br>Intervention component*: 10 | 1. Awareness of MHP concept 2. Average fruit consumption per day 3. Average vegetable consumption per day 4. Adequacy of FV intake | MHP awareness is associated with adequate FV intake among Malaysian adults with abdominal obesity, diabetes mellitus, hypertension, and hypercholesterolemia. |
| **Chua SS (2012) [48]** Pre/Post study | T2DM/HPT/ Dyslipidemia (n = 477) Study location(s): Klang Valley | Pharmacist counselling (a component of Community-Based Cardiovascular Risk Factors Intervention Strategies (CORFIS)) • 6 months | - The study involved a group of pharmacists, dietitians and nurses. Each participant was interviewed and counselled by one of the pharmacists, dietitians and nurses at the GP clinic. - Each participant spent about 30–60 min with each of the healthcare providers. - During the 24-week follow-up assessments, the pharmacist reviewed the participants' medications and counselled the participants every 4 weeks, noted any Pharmaceutical Care Issues (PCIs) encountered by the participants and helped to resolve the PCIs. - If required, the pharmacist would contact the GP concerned to alert, discuss and if possible, to resolve the PCIs which could affect the participants' clinical outcomes. - The dietitians provided dietary advice while the nurses advised the patients on general healthcare such as foot care.<br><br>*Intervention component*: 2, 3, 8, 9, 14* | 1.Types of PCIs encountered by the participants 2.Causes of PCIs 3.Clinical significance of PCIs 4.Outcome of the interventions made by the pharmacists | 1.53.7% had at least one PCI, with a total of 706 PCIs. 2.These PCIs included drug-use problems (33.3%), insufficient awareness and knowledge about disease conditions and medication (20.4%), adverse drug reactions (15.6%), therapeutic failure (13.9%), drug-choice problems (9.5%) and dosing problems (3.4%). |

*(Continued)*

**Table 2.** (Continued)

| First author (publication year) & Study Design | Study population (n) & study location (state(s)) | Intervention name & intervention period | Brief description of the intervention & intervention component* | Outcome measures | Results (Primary Outcome) |
|---|---|---|---|---|---|
| **HCP (n = 4)** | | | | | |
| **Tay CL (2022) [49]** Quasi-experimental study | HCPs (n = 30) Study location(s): Perak | Multi-faceted intervention to increase ED screening • One meeting & one-day workshop | - An audit and feedback and a mandate from management <br> − 1-day workshop on men's sexual health included short lectures on ED screening, assessment and management, case scenarios, role play, group discussions and presentations, and interactive question-and-answer sessions before and after the workshop. <br><br> **Intervention component*: 1, 6** | 1. ED knowledge score 2. Confidence level on the screening of ED 3. Screening and detection rate of ED | 1.The mean knowledge score was significantly increased before and after the workshop. 2.The median confidence level increased significantly after the workshop. 3.After the audit and feedback with mandate from management (phase 1) on 10 July 2019, the ED screening rate increased from 15% in July 2019 to 18.9% in September 2019. After the interactive face-to-face workshop (phase 2) in Sept 2019, the ED screening rate was maintained (Oct-Dec 2019). The ED screening rate subsequently decreased from January to February 2020. 4.There was a 4–5-fold increase in the detection rate of ED after phase 1, and it remained static after the phase 2 intervention. |
| **Wong WJ (2020) [50]** Quasi-experimental study | HCPs (n = 2257) Study location(s): Selangor/ Johor | Enhanced Primary Healthcare (EnPHC) • 17 months | EnPHC interventions include: - Integrated specialised services (ISS) - Cardiovascular care bundle medication therapy adherence counselling - Clinical and prescribing audits - Primary triage counter - Secondary triage counter - Care Coordinators - Family Health Team (FHTs) - NCD care form - Identification of medication refill defaulter <br><br> **Intervention component*: 3, 4, 5, 6, 9, 10, 14** | 1. Job satisfaction questionnaire (6 questions) | 1.At post-intervention, HCPs from the intervention group reported higher levels of stress 2.No statistically significant differences for the remaining five items (work does not make sense, interest with work, overloaded with administrative detail, well-respected job, balance between effort and reward) |
| **Lim SC (2020) [51]** Quasi-experimental, mixed methods study | HCPs (n = 77) Study location(s): Selangor/ Kuala Lumpur | Steno REACH Certificate Course (SRCC) • 6 months | SRCC curriculum had ten modules and was designed to include independent online study and face-to-face classroom time <br><br> **Intervention component*: 1** | 1. Clinical diabetes-related knowledge 2. Technical clinical skills and soft skills (patient engagement, empowerment and communication) via OSCE and direct clinical observation 3. Diabetes-related attitudes using DAS-3 | 1.For both physicians and nurses, mean post-test scores for the MCQ (clinical diabetes-related knowledge) and OSCE were higher in the intervention groups (Arms 1 and 2) compared to the control groups (Arms 3 and 4). 2.Both clinical and soft skills significantly improved for physicians and nurses. 3.No differences were observed for any of the five DAS-3 subscales (need for special training in education; seriousness of T2D, overall value of tight glucose control in diabetes care, psychosocial impact of diabetes on patients, and patient autonomy) between the intervention and control arms for either physicians or nurses. |
| **Bondi ME (2020) [52]** Quasi-experimental study | HCPs (n = 82) Study location(s): Sabah | Educational intervention module (wound) • 2 days | The educational intervention module was developed, and adapted from the Wound Care Guidelines consisted of three main sections: -Basic wound principle -Concept of wound care management -Principle aspect of wound care management. <br><br> **Intervention component*: 1** | 1. Knowledge, practice and attitude towards wound care management | 1.The intervention group participants had a high level of knowledge as compared to the control group. 2.There was a tremendous change of attitude level in the intervention group (100%), while only 38.5% of the participants in the control group had a positive level of attitude. 3.A higher practice of the participants in the intervention group while the level of practice in the control group remained unchanged at moderate practice and none in high level of practice. 4.There were significant within subject and between subject effects in the level of knowledge, attitude and practice over time. |

*Types of intervention: 1 = Education; 2 = Self-monitoring (glucose or blood pressure); 3 = Family doctor concept; 4 = Medication Therapy Adherence Clinic (MTAC); 5 = Care Coordinator; 6 = Audits & Feedback; 7 = Screening program; 8 = Regular follow-up ± longer consultation time; 9 = Multidisciplinary team; 10 = Decision aid (including guidelines); 11 = Nutrition therapy; 12 = Social Support; 13 = Dosing Time; 14 = Patient management system.

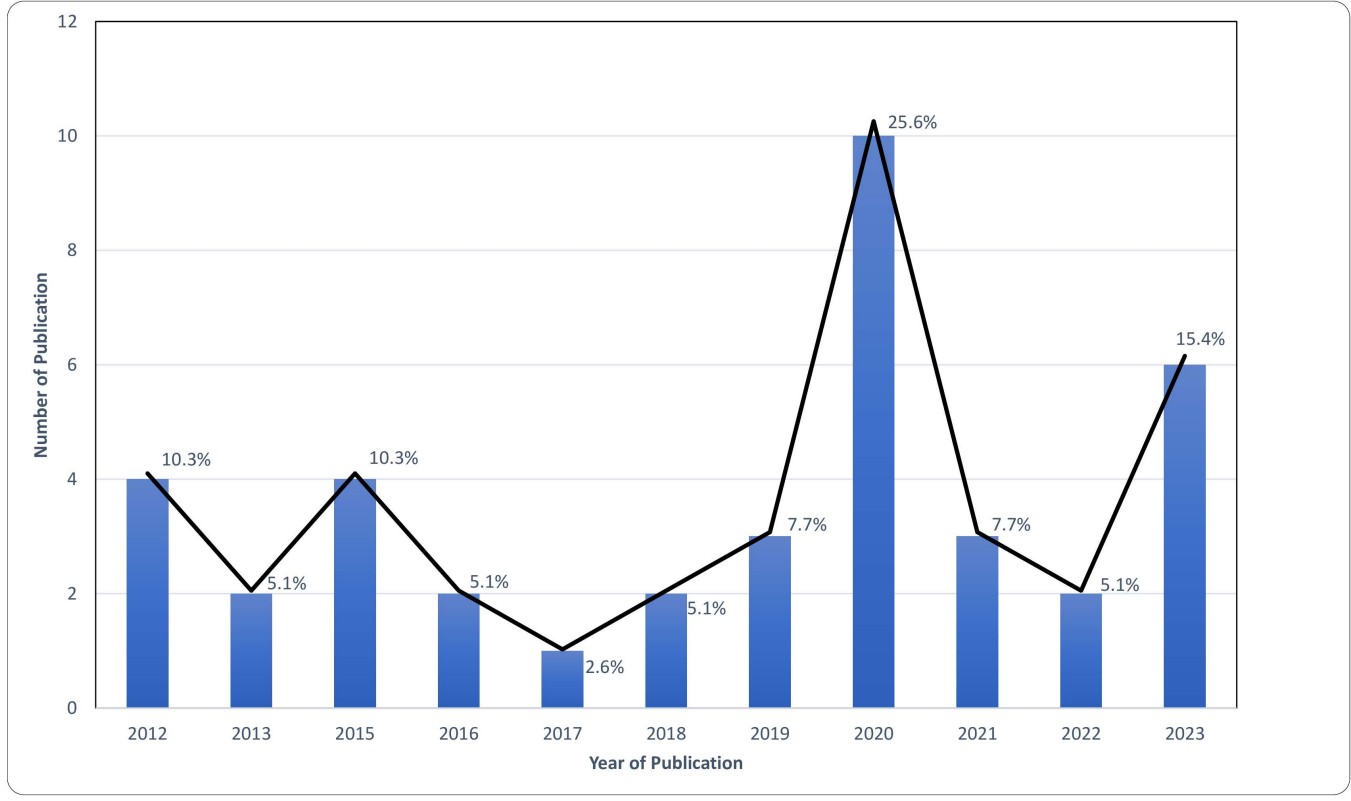

**Fig 2. Publications Trend (by year).**

**(ii) Decision Aid (n=11).** The decision aid intervention component can be categorised into two groups: those supporting patients and those supporting providers. Patient-focused decision aids were designed to support patient self-management, enabling individuals to monitor their health and make informed decisions. Examples include the Global Cardiovascular Risks Self-Management Booklet [26,27], food pyramid charts and T2DM booklets [38], and visual aids such as colour-coded HbA1c graphs [41]. In contrast, provider-focused decision aids aimed to standardise care delivery and guide clinical decision-making. These include the Integrated Care Pathways and NCD care form used in the EnPHC intervention [14,50], the Simpler tool for structured counselling [29,30], the CPG quick reference in EMPOWER-PAR intervention [26,27] and prompt sheets to facilitate sensitive consultations [15].

**(iii) Self-monitoring (n=6).** The self-monitoring component was categorized into technology-assisted and conventional approaches. Technology-assisted self-monitoring involved the use of digital tools such as a gluco-telemeter [28] and a web-enabled glucometer with automated feedback triggered by consecutive out-of-range readings [37]. Conventional self-monitoring approaches include providing patients with a glucometer with reagent test strips [19], post-prandial glucose monitoring [40] and loaning BP monitors for BP monitoring at home [46,48].

**(iv) Personalized Care Model (n=6) & Multidisciplinary Team (n=6).** The personalized care models were operationalized through the Family Health Team (FHT) in the EnPHC intervention, the FDC, and the Community-Based Cardiovascular Risk Factors Intervention Strategies (CORFIS) program, with the aim of improving continuity and comprehensiveness of care. These models emphasized continuity of care by the same provider or team. All these studies combined personalized care models with a multidisciplinary team, which refers to involvement of allied health professionals, to deliver comprehensive care [14,26,27,46].

*(v) Regular follow-up and longer consultation time (n=6).* The frequency and duration of in-person follow-ups varied across studies. For example, Tajudin TR (2020) included monthly follow-ups over six months, with each session lasting at least 30 minutes [25]; Ismail M (2013) scheduled doctor visits every two months [39]; and the CORFIS intervention included monthly counselling by trained allied health professionals [46]. In contrast, telephone follow-ups did not involve doctors. These were carried out monthly by pharmacists for prescription refill reminders and medication reassessment [38] or bi-weekly by trained nurse educators [43].

*(vi) Medication therapy adherence clinic (MTAC) (n=5) & Care Coordinator (n=2).* MTAC components can be diabetes-specific or cardiovascular care-specific. MTAC was done through pharmacist medication reviews and patient adherence monitoring. An additional role – Care Coordinator was implemented in EnPHC [14,50], which involved linking patients with FHTs and coordinating with healthcare providers to manage patient status and referrals.

*(vii) Patient management system (n=4).* Several studies incorporated the patient management system component, such as the visit checklist in the EnPHC intervention [14,50] and the web-based application in the CORFIS intervention [46,48], to capture patient data, monitor appointments and laboratory results, and coordinate care among providers.

*(viii) Audits and feedback (n=3).* The audits and feedback component tracked performance through regular reports and additional monitoring indicators [14,49,50]. Both the Care Coordinator component and audits and feedback component did not show improvement in glycemic control.

*(ix) Social support (n=3).* The social support component was reported in three studies, including a peer- and community-based approach with peer mentors [36], a faith-based intervention involving Quran recitation [44], as well as counselling and motivational interviewing to enhance patient engagement [34].

*(x) Screening program (n=2).* Two studies included screening programs: free blood tests during diabetes awareness day [25] and eye examinations to screen for retinal disease [24].

*(xi) Nutrition therapy (n=1) and dosing time (n=1).* One study prescribed structured low-calorie meal plans [34]. Another examined the optimal timing for simvastatin administration, a lipid-lowering drug, comparing doses taken after breakfast, after dinner, or at bedtime [42].

## Summary of interventions based on chronic care model

S3 Table showed the summary of the intervention characteristics based on duration, intervention target level and CCM domains. Out of the 32 interventions, 27 of them targeted the patient level, either alone or in combination with provider or system levels (S3 Table). When we matched the interventions to the CCM domains, we found that most of the interventions involved the following two domains: self-management support (n = 24) and delivery system design (n = 21). Community linkage is the domain that was least covered (n = 3). There was no intervention that covers all the CCM domains. Further details are provided in S4 Table.

**Chronic care models domains.**

(i) Self-management support (n=24)

Twenty-nine articles included 24 interventions that targeted the self-management support domain. The self-management components usually involve education and continuous follow-up. There was involvement of a multidisciplinary team in some studies, such as trained nurse educators [43], allied health support [21,46,48], and pharmacists-led medication adherence therapy clinic (MTAC) [14,50] or home-based education. Supporting materials such as prompt sheets, flip charts, information booklets, glucometers and measuring spoons were also used. Patients were involved in discussion and goal setting.

(ii) Delivery System Design (n=21)

Twenty-one interventions, reported across 28 articles, addressed the delivery system design domain of the CCM. Most interventions involved creating a multidisciplinary team for disease management or increasing the frequency of follow-ups

or monitoring. Two interventions, namely the FDC and EnPHC, ensure personalised care by assigning patients and populations from the same geographical area to the same primary healthcare team. Some interventions include regular monitoring of patients' glucose levels using telemonitoring where glucose readings are transmitted to a central server and the participants receive automated feedback on their results [28], or monthly reporting of SMBG results to the nurse [39]. In diabetes MTAC, pharmacists are granted authorization by the physicians to adjust insulin doses and actively participate in recommending medication regimens and laboratory investigations.

(iii) Decision support (n=8)

Eight interventions, reported across 10 articles, addressed the CCM decision support domain. Interventions that aim to support healthcare providers in their decision-making process, include improving access to clinical practice guidelines, such as by using QR code, agreed protocols, or integrated flow charts for different diseases [14,26,27,46,50]. Besides that, there are workshops or continuous medical education that aim to improve providers' knowledge and skills [49,51,52] or tools (e.g., flipchart, healthy plate portion guide) as a reminder/guide [15,47].

(iv) Clinical information system (n=7)

Seven interventions matched the CCM clinical information system domain. An automated feedback mechanism for patient glucose monitoring was incorporated in two studies [28,37]. There was also the use of a web-based application [46] or a visit checklist [14,50] to capture patient data and coordinate patient care. Besides, reminders were performed using either text messages [45] or telephone [38].

(v) Health System Organization (n=13)

Thirteen interventions aligned with the CCM health system organisation domain. These included changes in healthcare team structure or function, such as the formation of dedicated teams for specific zones like FHT or FDC [14,22,23,50] and multidisciplinary chronic disease management teams [26,27]. MTAC services enabled pharmacists to closely follow up with patients and adjust dosages as needed [14,17–19,50]. Other interventions include establishing a designated diabetic clinic [25] and ophthalmology team screening [24]. Apart from the team structure changes, some interventions increased the frequency of monitoring through telemonitoring [28], SMBG [39] and biweekly group sessions [32,33]. Other interventions include home visits [38], group counselling [45] and change of key performance indicators to improve screening [49].

(vi) Community Linkages (n=3)

There were three interventions that focused on the community linkages domain. First, in the CORFIS study, patients were provided information about local patient associations and support groups [46]. The Malaysia Healthy Plate (MHP) was a public health intervention to the community by showing the tips about portion size in a plate [47]. Besides that, the diabetes awareness day which offers free blood tests, diabetes meals and talks was also an intervention that fits the CCM community linkage domain [25].

**Outcome measures**

The most frequently reported outcome measures were the intermediate clinical indicators, including HbA1c (n = 17), blood pressure (n = 12) and cholesterol levels (n = 12). Knowledge, attitude and practice were also commonly measured (n = 10). Other outcome measures were medication adherence, quality of life, patient or provider satisfaction, processes of care, prescription, and emotional distress. A comprehensive list of outcomes is provided in Table 2.

**Discussion**

This scoping review provides an overview of PHC interventions in Malaysia targeting the management of diabetes, hypertension, and/or dyslipidaemia. Considering the rising prevalence of NCDs, it is encouraging to observe an increase in the

number of interventions implemented and evaluated in recent years. This positive trend may reflect growing awareness among healthcare stakeholders, increased investment in NCD programs, and alignment with both national and global priorities for NCD prevention and control. Notably, however, many of the interventions focused on diabetes management, with comparatively few interventions addressing hypertension and/or dyslipidaemia. This imbalance is concerning given the high prevalence of hypertension and dyslipidaemia in Malaysia. Future research and intervention efforts should be more evenly distributed to ensure that these conditions receive adequate attention within the PHC setting.

This scoping review identified an uneven geographic distribution of study locations, with a concentration of research particularly in the states of Selangor, Kuala Lumpur, Kelantan, and Johor. Selangor and Kuala Lumpur, being more urbanized and better resourced, likely benefit from stronger healthcare research infrastructure, established academic institutions, and greater access to funding and skilled human resources. Another plausible explanation is that in states with higher prevalence, local health authorities or organizations may have already implemented interventions or programs without formal evaluation or dissemination through peer-reviewed publications, hence not captured in this scoping review. These efforts, while potentially impactful, remain undocumented in the academic literature and therefore contribute little to the national or global evidence base. On the other hand, as the current research focuses more in the urban areas, the unique challenges of managing these three diseases in more rural areas might be overlooked. The lifestyle and environmental factors as well as healthcare accessibility might differ significantly from the urban setting. The geographical imbalance highlights the need for deliberate strategies to enhance research activities, intervention implementation and evaluation in less-studied states, particularly those with high disease burdens and unique local needs. Addressing this gap is essential to ensure that research findings and interventions are representative and relevant across Malaysia's diverse regions and populations.

Many studies in this review had relatively small sample sizes, which may limit the statistical power and generalisability, particularly in Malaysia's diverse population where ethnicity, culture, socioeconomic status, and geography influence health behaviours and outcomes. In our review, we found two interventions which are culturally sensitive – the use of Quran recitation audio [44] and Malaysian trans-cultural diabetes nutrition algorithm (tDNA) [34]. From literature, ethnic differences were observed in diabetes control and complication rates, with Chinese patients shown to experience more complications despite having better glycemic control, while Indian patients have been found to have higher rates of nephropathy [53]. Additionally, gender differences influence the factors affecting blood pressure control [54], and also modify the impact of age on glycemic control [55]. A review of hypertension management also emphasized the impact of cultural practices and health beliefs on a patient's adherence to treatment, highlighting the need for culturally tailored interventions rather than a single, universal strategy [56]. Thus, the "one-size-fits-all" approach is not ideal for NCD management, especially with diabetes, hypertension and dyslipidemia. It is important to consider these sociocultural differences, especially when delivering lifestyle modification interventions and providing educational information. National NCD programs should incorporate sociocultural considerations into prevention and management strategies, develop targeted interventions for high-risk subgroups, and strengthen health communication that is sensitive to cultural and gender-specific determinants to achieve equitable and sustainable impact.

In addition, about half of the interventions identified in this scoping review were implemented over relatively short durations (between one to six months), thereby raising questions regarding their long-term sustainability, scalability, and feasibility for nationwide implementation. Interventions such as EnPHC, Malaysia Health Plate, FDC and DMTAC were designed for nationwide implementation, whereas the others are short term programs. In order for the intervention to be adopted widely as public health programs, despite effectiveness, sustainability and scalability need to be considered [57]. Future interventions should integrate implementation research principles to directly address the requirements for effective, sustainable, and adaptable for nationwide deployment, particularly considering cultural and setting differences across the Malaysian PHC system.

The diversity of intervention components observed in this review reflects the complex interplay of medical, lifestyle, and psychosocial factors in NCD care [58]. The predominance of education and self-monitoring components in these

Malaysian interventions aligns with international standards, such as the American Diabetes Association's Standard of Care 2025 and a Consensus Report for Diabetes Self-Management Education and Support (DSMES), highlighting the need to have DSMES not only at diagnosis, but also during major life transitions and the onset of complications [59,60]. The educational content in these interventions cover a broad range of topics and targeted at both patients and providers. However, none of the reported interventions addressed all seven domains of the ADCES7 Self-Care Behaviours™ (ADCES7) framework. Future programs could incorporate higher-level domains such as "problem solving" and "reducing risks" to comprehensively cover self-care topics [61]. Furthermore, most studies that incorporated education and self-monitoring components focused on immediate outcomes, such as knowledge improvement rather than intermediate clinical outcomes like glycemic control or cardiovascular risk reduction. Furthermore, they were conducted over relatively short durations. These findings highlighted the need to evaluate both the long-term effectiveness and the quality of DSMES delivery to ensure sustained patient engagement and meaningful clinical impact.

Personalized care models enhance continuity, foster patient-provider trust, and improve coordination, while multidisciplinary teams bring together diverse expertise to address the medical, nutritional, and psychosocial needs of people with diabetes. Both align closely with the delivery system design and health system organization domains of the CCM, which emphasizes proactive, patient-centred, and team-based care. However, only two interventions in this review (EnPHC and CORFIS) employed a combination of both components, suggesting a gap as well as an opportunity for advancement of diabetes, hypertension and dyslipidemia management in Malaysian PHC. The barriers to wider adoption of these models include workforce shortages, particularly the limited number of family medicine specialists, diabetes educators, physiotherapists, and dieticians, along with competing demands of multiple parallel programs within the clinic, high staff turnover, and limited interprofessional training [62,63]. Given the variability in structure, manpower and equipment across clinics, strong and committed leadership together with innovative strategies such as structured task-shifting protocols, protected time for team-based care discussions, integrated one-stop diabetes services and leverage on technology, are needed to integrate personalized care models and multidisciplinary teams into routine PHC practice [63].

Community linkage was the least targeted CCM domain in this review. A similar trend was observed in other reviews that included studies from various countries [64,65]. In this present review, the interventions addressing community linkage consisted of public awareness events with diabetes screening, health campaigns like the Malaysia Healthy Plate, and provision of information on local support groups. These activities, while valuable, fall short of establishing a sustained, structured partnership between healthcare teams and community organizations. The experiences from other countries demonstrated that robust and high-impact community linkage requires dedicated, continuous mechanisms, such as formalized collaborations between academic centres, physicians, and non-governmental organizations [66], or the utilization of trained Community Health Workers to provide in-home support and counselling [67]. This review underscores the urgent need to create more robust community linkages in Malaysia that can extend the reach of clinic-based care, reinforce self-management, and provide social and emotional support to patients.

This scoping review's strength is the systematic mapping of interventions to the six CCM domains and a detailed breakdown of intervention components, which offers insights into strategies that were used in Malaysian PHC. By presenting the interventions by component, this review provides a practical reference for policymakers and practitioners seeking to adapt similar approaches. We also uncover research gaps in hypertension and dyslipidaemia, as well as community linkage for NCD management. This information can serve as a guide for future research opportunities in Malaysia. Besides that, our broad search strategy, which included a local database (MyMedR), no language restrictions, and snowballing, helped to maximise the inclusion of relevant studies. However, we have several limitations, that includes the absence of formal quality assessment or critical appraisal of included studies, as our focus was on describing interventions and mapping them to the CCM domains. The scoping nature of the review prevents conclusions about intervention effectiveness to be drawn, as outcomes were not systematically synthesized or compared. Besides that, potential publication bias was inherent in this review as interventions implemented but not published in peer-reviewed literature may have been missed.

## Conclusion

This review reveals a significant imbalance in Malaysia's PHC interventions for NCDs; while self-management and clinical systems are well-addressed, a heavy focus on diabetes and a critical neglect of community linkages leave major gaps in managing the high burden of these three diseases in Malaysia. The challenge for Malaysian policymakers is not only to scale effective and culturally tailored interventions but to strategically invest in community partnerships as well as broaden research focus beyond diabetes and urban areas. Closing these gaps is essential for creating a sustainable and equitable PHC system capable of tackling the full spectrum of NCDs.

## Supporting information

**S1 Table. Preferred Reporting Items for Systematic reviews and Meta-Analyses extension for Scoping Reviews (PRISMA-ScR) Checklist.**
(DOCX)

**S2 Table. Search strategies.**
(DOCX)

**S3 Table. Characteristics of interventions\* (n = 32).**
(DOCX)

**S4 Table. Descriptions of the interventions in terms of intervention levels and CCM domains.**
(DOCX)

## Acknowledgments

We would like to thank the Director General of Health Malaysia for his permission to publish this article.

## Author contributions

**Conceptualization:** Xin Rou Teh, Swee Hung Ang, Pei Jia Lee, Azah Abdul Samad.

**Data curation:** Xin Rou Teh, Pei Jia Lee, Muhammad Izzuddin Mohd Ropidi.

**Formal analysis:** Xin Rou Teh, Pei Jia Lee.

**Funding acquisition:** Swee Hung Ang.

**Investigation:** Muhammad Izzuddin Mohd Ropidi.

**Methodology:** Xin Rou Teh, Swee Hung Ang.

**Project administration:** Xin Rou Teh.

**Supervision:** Kim Sui Wan.

**Validation:** Swee Hung Ang, Muhammad Izzuddin Mohd Ropidi, Azah Abdul Samad, Sheamini Sivasampu, Kim Sui Wan.

**Visualization:** Xin Rou Teh.

**Writing – original draft:** Xin Rou Teh.

**Writing – review & editing:** Xin Rou Teh, Swee Hung Ang, Pei Jia Lee, Azah Abdul Samad, Sheamini Sivasampu, Kim Sui Wan.

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
