## [Decision Letter · Decision Letter 0]

3 Feb 2026

Dear Dr. Teh,

Thank you for submitting your manuscript to PLOS ONE. After careful consideration, we feel that it has merit but does not fully meet PLOS ONE’s publication criteria as it currently stands. Therefore, we invite you to submit a revised version of the manuscript that addresses the points raised during the review process.

We look forward to receiving your revised manuscript.

Kind regards,

Mohammed Abutaleb, PhD

Academic Editor

PLOS One

“The funding of this study was supported by a Malaysia Research Grant from the Ministry of Health Malaysia.”

“The funding of this study was supported by a Malaysia Research Grant from the Ministry of Health Malaysia (NMRR-24-00853-YHT).”

“The funding of this study was supported by a Malaysia Research Grant from the Ministry of Health Malaysia.”

Additional Editor Comments:

work on the Reviewer 2 comments

Reviewers' comments:

Reviewer's Responses to Questions

**Comments to the Author**

1. Is the manuscript technically sound, and do the data support the conclusions?

Reviewer #1: Yes

Reviewer #2: Partly

Reviewer #3: Yes

2. Has the statistical analysis been performed appropriately and rigorously?

Reviewer #1: Yes

Reviewer #2: No

Reviewer #3: N/A

3. Have the authors made all data underlying the findings in their manuscript fully available?

Reviewer #1: Yes

Reviewer #2: Yes

Reviewer #3: Yes

4. Is the manuscript presented in an intelligible fashion and written in standard English?

Reviewer #1: Yes

Reviewer #2: No

Reviewer #3: Yes

Reviewer #1: This is an excellent and timely review. Given the high rates of NCDs in Malaysia, mapping these interventions against the Chronic Care Model is very valuable.

The paper is well-written and clear. I particularly appreciate how you highlighted the specific gaps—such as the over-focus on diabetes and the lack of community linkages. This will be very useful for future policymakers and researchers.

I have no changes to request. Congratulations on a solid piece of work.

Reviewer #2: The authours have attepted a topic of importance and they need to be coended for that. But having said that, it is presented in a very confusing and shabby manner. THe table sumarising the studies and the interventions are too large and confusing.

THE main objective namely, the relationship of the interventions individually to the coponents of the primary care model is not presented cllearly. It is hard to read and understand what they are trying to say.

Reviewer #3: This scoping review article by Teh et al. was well written overall. It is clear that this review set out to describe the primary health interventions for diabetes, hypertension and/or dyslipidemia. The authors were successful in articulating the rationale for the need for this scoping review, provided supplementary data including a summary of the intervention against the Chronic Care Model.

.

Reviewer #1: **Yes:**Ganesh Praneeth Roy AvulaGanesh Praneeth Roy AvulaGanesh Praneeth Roy AvulaGanesh Praneeth Roy Avula

Reviewer #2: No

Reviewer #3: **Yes:**Veronica SegbedzieVeronica SegbedzieVeronica SegbedzieVeronica Segbedzie

---

## [Author Response · Author response to Decision Letter 1]

23 Feb 2026

Reviewer #2 comments:

The authours have attepted a topic of importance and they need to be coended for that. But having said that, it is presented in a very confusing and shabby manner. THe table sumarising the studies and the interventions are too large and confusing.

THE main objective namely, the relationship of the interventions individually to the coponents of the primary care model is not presented cllearly. It is hard to read and understand what they are trying to say

Feedback from authors:

Thank you for your comment. We have revised Table 2 to improve its clarity and presentation. Specifically, we have reduced the number of columns to make it more concise and easier to read. In addition, we have edited the Results column to enhance readability and ensure the findings are presented more clearly.

One of the objectives of the study is to summarise the intervention components, where it is now shown in Table 2, together with the description of the interventions.

---

## [Editor Report · Decision Letter 1]

25 Mar 2026

Primary Health Care Interventions Targeting Diabetes, Hypertension or Dyslipidemia in Malaysia: A Scoping Review

PONE-D-25-65577R1

Dear Dr. Xin Rou Teh,

We’re pleased to inform you that your manuscript has been judged scientifically suitable for publication and will be formally accepted for publication once it meets all outstanding technical requirements.

Kind regards,

Mohammed Abutaleb, PhD

Academic Editor

PLOS One
---

## [Editor Report · Acceptance letter]

PONE-D-25-65577R1

PLOS One

Dear Dr. Teh,

I'm pleased to inform you that your manuscript has been deemed suitable for publication in PLOS One. Congratulations! Your manuscript is now being handed over to our production team.

Kind regards,

on behalf of

Dr. Mohammed Abutaleb

Academic Editor

PLOS One